# Hazel Leaf Polyphenol Extract Alleviated Cisplatin-Induced Acute Kidney Injury by Reducing Ferroptosis through Inhibiting Hippo Signaling

**DOI:** 10.3390/molecules29081729

**Published:** 2024-04-11

**Authors:** Mingyang Sun, He Chang, Fangyang Jiang, Wenjing Zhang, Qingxuan Yang, Xinhe Wang, Guangfu Lv, He Lin, Haoming Luo, Zhe Lin, Yuchen Wang

**Affiliations:** College of Pharmacy, Changchun University of Chinese Medicine, Changchun 130117, China; sunmy0421@163.com (M.S.); 15843593185@163.com (H.C.); fangyangsun@163.com (F.J.); 15022164457@163.com (W.Z.); qingxuany@163.com (Q.Y.); m17704303262_1@163.com (X.W.); lvgf@ccucm.edu.cn (G.L.); linhe@ccucm.edu.cn (H.L.)

**Keywords:** hazel leaf, phenolics, oxidative stress, ferroptosis, Hippo pathway

## Abstract

Derived from hazelnuts, hazel leaf has been utilized in traditional folk medicine for centuries in countries such as Portugal, Sweden, and Iran. In our previous investigations, we conducted a preliminary assessment of the hazel leaf polyphenol extract (referred to as ZP) and identified nine compounds, such as kaempferol and chlorogenic acid, in its composition. ZP has shown promising properties as an antioxidant and anti-inflammatory agent. Our research has revealed that ZP has protective effects against cisplatin-induced acute kidney injury (AKI). We conducted a comprehensive examination of both the pathological and ultrastructural aspects and found that ZP effectively ameliorated renal tissue lesions and mitigated mitochondrial damage. Moreover, ZP significantly suppressed malondialdehyde levels while increasing glutathione and catalase concentrations in the kidneys of AKI-induced mice. ZP decreased the number of apoptotic cells and decreased pro-apoptotic protein expression in the kidneys of mice and human renal tubular epithelial cells (HK-2). Furthermore, treatment with ZP increased the levels of proteins marking anti-ferroptosis, such as GPX4, FTH1, and FSP1, in experiments both in vivo and in vitro. We elucidated the underlying mechanisms of ZP’s actions, revealing its inhibitory effect on Yap phosphorylation and its regulation of Lats expression, which exert a protective influence on the kidneys. Furthermore, we found that inhibiting the Hippo pathway compromised ZP’s nephroprotective effects in both in vitro and in vivo studies. In summary, this research shows that ZP exhibits renoprotective properties, effectively reducing oxidative damage, apoptosis, and ferroptosis in the kidneys by targeting the Hippo pathway.

## 1. Introduction

Acute kidney injury (AKI) is a prevalent and increasingly common condition worldwide [1]. Acute kidney injury (AKI) is characterized by a swift decline in renal function, leading to injury and the demise of renal tubular cells, particularly those in the proximal tubules. This can potentially result in the development of chronic kidney disease (CKD) [2]. Nephrotoxic drugs, sepsis, and ischemia–reperfusion (I/R) are the main causes of AKI [3]. Cisplatin, a commonly used and effective antitumor medication, is often hindered in its use due to nephrotoxicity. This renal toxicity predominantly impacts the kidney by triggering proximal tubular absorption, resulting in intracellular cisplatin levels up to five times greater than those found in the bloodstream [4]. AKI, induced by cisplatin (CDDP), triggers inflammation, oxidative stress, vascular damage, endoplasmic reticulum stress, cellular necrosis, and apoptosis [5].

Abnormal oxidative stress is one of the primary reasons why CDDP promotes AKI. CDDP can cause damage to intracellular mitochondrial DNA, decrease levels of glutathione (GSH) and superoxide dismutase (SOD), and promote the production of reactive oxygen species (ROS), leading to oxidative stress. Ferroptosis, characterized by iron and lipid peroxidation accumulation, is a type of cell death linked to ROS build-up. Recent research has demonstrated the significant involvement of ferroptosis in AKI [6]. Ferroptosis is a unique form of ‘regulated cell death’ that relies on iron and is characterized by lipid peroxidation, as well as changes in the mitochondrial structure and cristae [5]. Substantial evidence indicates that it is crucial to focus on ferroptosis to mitigate damage to tissues. Key controllers of ferroptosis include glutathione peroxidase 4 (GPX4) and the cystine/glutamate antiporter SLC7A11 (referred to as xCT) [7]. When GPX4 is absent, acute kidney injury (AKI) arises spontaneously as a result of damage to renal tubules; however, when GPX4 is increased, this damage is lessened [8]. Furthermore, the breakdown of GPX4 via chaperone-mediated autophagy enhances the process of iron collapse in renal tubular cells during AKI [9]. Subsequently, Hu et al. demonstrated that inhibiting ferroptosis with the compound ferrostatin-1 (Fer-1) through an intraperitoneal injection led to a significant reduction in kidney tissue cell death in mice with AKI, suggesting the involvement of ferroptosis in the development of AKI [10]. Hence, it is evident that apoptosis, ferroptosis, and cellular damage caused by free radicals have significant impacts on the development of AKI.

The Hippo pathway plays a crucial role in regulating cell proliferation, differentiation, and apoptosis [11]. The yes-associated protein (YAP), the principal effector molecule downstream in this pathway, is closely linked to the progression of AKI. Research has shown that YAP is abundantly expressed and predominantly found in the nucleus in both AKI patients and mice experiencing ischemia–reperfusion-induced AKI [12,13]. Inhibiting YAP activation or knocking out YAP has been shown to alleviate AKI kidney disease and improve functional impairment and subsequent repair [14]. Research on the impact of the Hippo/YAP signaling pathway on the regulation of the transferrin receptor (TFRC) and ferritin light chain (FTL) in the context of ferroptosis has shown interesting findings. Specifically, elevated levels of YAP have been linked to the upregulation of TFRC, ultimately facilitating ferroptosis and resulting in the demise of cancer cells [15]. Nevertheless, there are instances where heightened YAP expression has been associated with a notable increase in FTL expression, leading to a decrease in ferroptosis and impeding the destruction of cancer cells [15]. The ECAD-NF2-Hippo-YAP signaling axis is involved in regulating the sensitivity of tumor cells to ferroptosis in mesothelioma, with ACSL4 identified as a key mediator [16]. However, the specific role and regulatory mechanism of YAP in renal ferroptosis remain unclear.

Hazel leaf, a significant by-product of hazelnuts, has been the subject of several studies [17]. These studies have revealed the presence of various compounds in hazel leaf extract, including diarylheptane, flavonol glycosides, and yangonin [18,19]. Another study focused on the aqueous extracts of three hazel leaves and identified eight phenolic compounds, such as caffeoyl tartaric acid, p-coumaroyl tartaric acid, and populinoid rhamnoside [20]. Our previous investigation also explored the composition of hazel leaves, revealing a total of 35 phenolic compounds in both their free and bound forms. These compounds included various phenolic acids, flavonoids, and catechins. The hazel leaf polyphenol extract (referred to as ZP) was found to contain nine key components, namely kaempferol, chlorogenic acid, myricetin, caffeic acid, p-coumaric acid, resveratrol, luteolin, gallic acid, and ellagic acid. The initial analysis showed that kaempferol had the highest concentration at 221.99 mg/g, while chlorogenic acid came in a close second at 8.23 mg/g [21]. Furthermore, after treating HUVEC cells with tert-butyl hydroperoxide (TBHP), it was found that phenolic extracts were able to reduce oxidative damage by enhancing SOD activity and decreasing the Malondialdehyde (MDA) content [21]. However, the molecular mechanisms and effects of ZP on AKI are still not well understood, particularly in relation to ferroptosis.

Within the scope of this investigation, we successfully created an in vitro and in vivo model of AKI by administering cisplatin to mice and human renal tubular epithelial cells (HK-2). In this study, we aimed to assess the potential kidney-protective properties of ZP by investigating its impact on oxidative stress, ferroptosis, and apoptosis. We conducted experiments both in vitro and in vivo to inhibit the expression of the Yap protein, which is a key player in the Hippo signaling pathway. By doing so, we sought to determine whether ZP exerted its nephroprotective effects through the modulation of this pathway. The results of our research are groundbreaking as they introduce novel strategies for mitigating ferroptosis, a phenomenon associated with kidney injury. Additionally, these findings offer a promising therapeutic approach for the treatment of AKI.

## 2. Results

### 2.1. ZP Relieves Cisplatin-Induced AKI

Our research examined the levels of renal function markers in mice, specifically blood urea nitrogen (BUN) and serum creatinine (SCr). Our findings indicate that ZP effectively diminished the elevation of BUN and SCr induced by cisplatin administration (Figure 1B,C) (*p* < 0.05). Additionally, we examined the histopathological and ultrastructural changes in the kidneys of mice using H&E staining and a transmission electron microscope (TEM). The findings are presented in Figure 1D,E. Following cisplatin administration, the epithelial cells of the mice renal tubules showed degeneration and some of them detached into the lumen. We also observed the presence of abnormal nuclei and the mild infiltration of inflammatory cells in the mice kidneys. However, when ZP was administered prior to treatment, it improved the observed pathological changes in the mice kidneys. The results from TEM demonstrated that ZP mitigated mitochondrial damage caused by cisplatin in the kidneys of mice. Moreover, the findings from an MTT assay revealed that the HK-2 cells’ proliferation ability in the ZP group showed a significant increase compared to the Model group (Figure 1F,G) (*p* < 0.05). Overall, these results indicate a beneficial role of ZP in mitigating cisplatin-induced AKI.

### 2.2. ZP Ameliorates Oxidative Damage in Mice Kidneys and HK-2 Cells

To investigate the impact of ZP on cisplatin-induced oxidative damage in mice kidneys, we analyzed the levels of SOD, MDA, glutathione (GSH), and catalase (CAT). Our findings suggest that cisplatin treatment led to a decrease in antioxidants, including GSH, SOD, and CAT, in the mice kidneys. The significant improvement was observed with the administration of ZP, as illustrated in Figure 2A–C (*p* < 0.05). Additionally, the analysis of MDA levels indicated a notable decrease in kidney samples from the ZP groups when compared to the Model group, as depicted in Figure 2D (*p* < 0.05). During our experimentation using HK-2 cells in the laboratory, the Model group showed a significant rise in ROS levels in comparison to the control group. Nevertheless, the application of ZP successfully suppressed the buildup of ROS (Figure 2E).

### 2.3. ZP Reduces Cisplatin-Induced Apoptosis in Mice Kidneys and HK-2 Cells

Mitochondrial damage is a key feature of apoptosis. The administration of ZP effectively improved mitochondrial injury caused by cisplatin. To assess the impact of ZP on cisplatin-induced apoptosis in renal cells, we performed TUNEL staining on mouse kidneys. The results showed a significant decrease (*p* < 0.05) in the number of apoptotic cells in the ZP groups compared to the Control group (Figure 3A). Furthermore, immunohistochemistry analysis revealed that cisplatin significantly increased the expression of Caspase 3 in mouse kidneys. However, this change was attenuated with ZP administration (Figure 3B). In vitro experiments using HK-2 cells also yielded similar results. The pro-apoptotic protein levels in HK-2 cells were assessed, and the results aligned with the in vivo tests. Specifically, ZP significantly reduced (*p* < 0.05) the protein expression of Caspase-9 and Caspase-3 induced by cisplatin (Figure 3C,D).

### 2.4. ZP Improves Cisplatin-Induced Ferroptosis in Mice Kidneys and HK-2 Cells

Lipid peroxidation is the primary factor responsible for triggering ferroptosis. In this study, we investigated the impact of ZP on ferroptosis by conducting immunohistochemistry to analyze the levels of representative ferroptosis proteins GPX4, ferroptosis suppressor protein 1 (FSP1), and ferritin heavy chain 1 (FTH1) in mice kidneys. The results showed a significant reduction in the positive area of GPX4, FSP1, and FTH1 in mice kidneys treated with cisplatin. However, the levels of these proteins were notably elevated in the ZP groups compared to the Model group (Figure 4A). In the same way, experiments conducted in vitro with HK-2 cells also revealed a significant rise in the protein levels of GPX4, FSP1, and FTH1 in the ZP cohorts when contrasted with the Model group (Figure 4B,C) (*p* < 0.05). These findings suggest that ZP effectively mitigates cisplatin-induced renal ferroptosis.

### 2.5. ZP Activates Hippo Signaling in Cisplatin-Treated Mice Kidneys and HK-2 Cells

To investigate the potential protective role of ZP on kidney function through the Hippo pathway, we conducted an immunofluorescence analysis to determine the levels of Yap. The results, shown in Figure 5A, demonstrated a significant decrease in Yap levels after cisplatin treatment. However, the ZP groups exhibited significantly higher levels of the protein compared to the Model group (Figure 5A). Furthermore, in AKI mice, we observed an upregulation of p-Yap and p-Lats expression, as well as a decrease in Yap expression (Figure 5B,C). Similar findings were observed in HK-2 cells, with increased levels of p-Yap, p-Lats, and decreased Yap expression (Figure 5D,E). Collectively, the immunofluorescent and protein expression data support the involvement of the Hippo signaling pathway in the kidney protective effects of ZP.

### 2.6. The Protective Effect of ZP on Mice Kidneys and HK-2 Cells Is Attenuated after YAP Phosphorylation

To validate the nephroprotective impact of ZP via Hippo signaling, a series of experiments were conducted in mice. The approach involved enhancing Yap phosphorylation using Dihydrexidine. The obtained results, which are presented in Figure 6, signify a considerable reduction in Yap levels within the kidneys of mice belonging to the ZP+ Dihydrexidine group in comparison to the ZP group (Figure 6A). Additionally, the ZP+ Dihydrexidine group manifested aggravated renal mitochondrial impairment, leading to the shedding of renal tubular epithelial cells and the infiltration of inflammatory cells (Figure 6B). The analysis of immunohistochemistry data exhibited a notable augmentation in GPX4 within the kidneys of mice belonging to the ZP+ Dihydrexidine group, contrasting with the ZP group (Figure 6B). Western blot analysis further revealed a significant upsurge in the phosphorylation levels of Hippo proteins in mice kidneys, including Yap and Lats, following treatment with Dihydrexidine in HK-2 cells (Figure 6C,D) (*p* < 0.05). Furthermore, the expression of anti-ferroptosis-related proteins, such as GPX4 and FTH1, was significantly diminished in the ZP+ Dihydrexidine group compared to the ZP group (Figure 6C,D) (*p* < 0.05). These findings propose that ZP hinders ferroptosis through Yap phosphorylation facilitated by the Hippo pathway.

## 3. Discussion

Cisplatin, also known as CDDP, is a commonly used chemotherapy medication. However, its use as an anti-cancer drug is limited due to its toxicity towards the kidneys and the potential risk of causing sudden kidney damage known as CI-AKI [22]. The occurrence of CI-AKI is closely associated with the accumulation of ROS and resulting oxidative stress. Additionally, damage to the mitochondria is also linked to CI-AKI. Another characteristic of CI-AKI is the presence of ferroptosis, a process involving mitochondrial dysfunction and harmful lipid peroxidation. Therefore, it is important to explore the use of natural antioxidants and medications that reduce ROS levels in the body to inhibit ferroptosis, providing potential treatment options for mitigating CI-AKI. ZP, a naturally occurring compound found in hazel leaf, has been documented to exhibit antioxidant characteristics. In this study, our objective was to evaluate the effect of ZP on AKI by conducting experiments on mice treated with cisplatin and HK-2 cells. Our focus was to understand the precise molecular mechanism underlying the effects of ZP. Our findings strongly indicate that ZP effectively modulates the Hippo signaling pathway, both in animal models and cell cultures, leading to a significant reduction in oxidative stress and ferroptosis. However, it is important to note that the renoprotective ability of ZP is noticeably diminished when Yap is phosphorylated, highlighting the crucial role of the Hippo pathway in ZP-mediated kidney protection.

The process of cell signaling is closely linked to intracellular ROS. The accumulation of ROS can disrupt the equilibrium between pro-oxidant and antioxidant reactions, leading to mitochondrial dysfunction. Excessive levels of ROS can result in cellular oxidative damage through the oxidation of proteins, lipids, and DNA [23]. The development of AKI induced by cisplatin is mainly due to the buildup of ROS and lipid peroxidation products, accompanied by decreased levels of antioxidant molecules such as SOD and GSH [24]. The findings of our research indicate that ZP exhibits protective effects on the kidneys in cases of cisplatin-induced AKI through its enhancement of the antioxidant system. ZP enhances the levels of SOD, GSH, and CAT, thereby decreasing the presence of ROS and lipid peroxidation products (such as MDA) in both renal tissues and HK-2 cells. Additionally, the accumulation of cisplatin in the mitochondria results in the production of significant levels of reactive oxygen species and the impairment of mitochondrial function. This in turn promotes mitochondrial permeability, leading to the release of pro-apoptotic factors and triggering cellular apoptosis. This indirectly enhances mitochondrial permeability, causing the release of pro-apoptotic factors and initiating cell apoptosis [25]. The Bcl-2 family protein acts as a crucial regulator of cell apoptosis through the mitochondrial pathway. The regulation of the pro-apoptotic protein Bax and the anti-apoptotic protein Bcl-2 is maintained by the cell. Caspase-3 activation is initiated by Bax, whereas Bcl-2 functions to inhibit apoptosis by blocking the movement of Bax to the mitochondria [26]. Consistent with previous studies, the results we discovered show that ZP effectively decreases the levels of pro-apoptotic molecules, such as Caspase-3 and Caspase-9, in the kidneys of mice and HK-2 cells. This suggests that ZP plays a crucial role in guarding against renal cell death.

Recently, studies have revealed the harmful effects of iron, specifically ferroptosis, which can lead to renal tissue damage [27,28,29]. Hu et al. conducted experiments using cisplatin-injected mice to create an AKI model and found that the ferroptosis inhibitor Fer-1 significantly reduced serum urea nitrogen and creatinine levels, thereby alleviating renal damage [10]. Another study by Zhao et al. supplemented sucrose iron in the cisplatin-induced AKI model and discovered that iron supplementation upregulated ferritin and ferroprotein, preventing the progression of AKI [30]. Although the signaling pathways of ferroptosis are not fully understood, it is known that the accumulation of unstable iron in cells, the generation of lipid peroxidation, and the impairment of mitochondrial function can all induce ferroptosis. Under pathological conditions, intracellular and external iron-death-related stimuli lead to an increase in the transferrin receptor TFR and a decrease in ferroportin (FPN), while promoting the release of free iron from intracellular ferritins including transferrin light chain (FTL) and transferrin heavy chain (FTH-1) [31]. Iron overload impacts mitochondrial function, leading to an elevated production of mitochondrial reactive oxygen species (mitoROS). The heightened oxidative stress induces hydroxyl radicals and various reactive oxygen species to attack phospholipid membranes, initiating lipid peroxidation and the buildup of lipid hydroperoxides, resulting in cell demise [32]. The unique structure of GPX4 enables it to interact with membrane phospholipids, making it a critical defense mechanism against ROS-mediated membrane peroxides [33]. GPx4, as an important antioxidant enzyme, uses glutathione (GSH) as a cofactor to convert lipid peroxides into non-toxic substances, thereby inhibiting ferroptosis [34]. Consequently, enhancing GPX4 activity can significantly mitigate ferroptosis. Moreover, the gene Fsp1 acts as a robust inhibitor of ferroptosis, functioning alongside the canonical GSH-dependent pathway of GPX4 [35]. In this study, we examined the mitochondrial morphology of mouse kidney cells and observed changes after cisplatin treatment. The mitochondria in kidney cells showed a significant decrease in size and a reduction or disappearance of the mitochondrial ridges. Additionally, we investigated the levels of GPX4, FTH1, and FSP1 in mouse kidneys using various methods including immunohistochemistry. We found a significant decrease in these proteins after cisplatin treatment, but ZP showed a significant improvement in these conditions. The in vitro experimental results were consistent with the in vivo experimental results, suggesting that ZP effectively prevents cisplatin-induced renal ferroptosis.

The Hippo pathway is a crucial regulator of cell proliferation, differentiation, and apoptosis, with YAP acting as its main downstream effector molecule [11]. In this pathway, upstream regulatory elements phosphorylate mammalian sterile-20-like kinase1/2 (Mst1/2), activating it. Phosphorylated Mst1/2 then phosphorylates and activates Lats1/2, which in turn directly phosphorylates the downstream protein YAP. Research has shown that phosphorylated YAP remains in the cytoplasm and cannot enter the nucleus, preventing its binding to the transcriptional enhanced associate domain (TEAD) and the initiation of downstream target gene transcription [36]. Therefore, the activation of the Hippo pathway leads to YAP phosphorylation and inactivation, while the inactivation of the pathway leads to YAP dephosphorylation, allowing it to enter the nucleus and bind to the TEAD transcription factor. This activation of downstream target gene expression contributes to the regulation of cell proliferation and the apoptotic process [16]. Kidney cells play a crucial role in regulating Hippo signaling in various situations such as cellular stress, ongoing pathological damage, harmful biomechanical signals, or cell malfunction [11]. AKI results in the temporary phosphorylation of YAP, which triggers the activation of genes and can impede the healing process. This hindrance can eventually cause the shift from AKI to Chronic Kidney Disease (CKD) and the progression of CKD [14,37]. Studies on ferroptosis-related research have observed that cells exhibit varying sensitivity to ferroptosis depending on their cell density and the regulation of YAP. At a high cell density, YAP activity decreases, leading to a decrease in cell sensitivity to ferroptosis. Conversely, at a low cell density, YAP activity increases, resulting in an increase in cell sensitivity to ferroptosis [38]. In hepatocellular carcinoma, the knockdown of AKR1C3 resulted in a decrease in YAP nuclear translocation, leading to the inhibition of cystine transporter SLC7A11. This inhibition caused an increase in the intracellular levels of ferrous iron, ultimately leading to ferroptosis [39]. Therefore, it is crucial to investigate how YAP regulates ferroptosis in cisplatin-induced AKI and the impact of ZP on the Hippo pathway. In this study, we observed an increase in the phosphorylation levels of Yap and Lats in the kidneys and HK-2 cells of AKI mice. We also found that ZP could inhibit this effect. However, when we enhanced the phosphorylation of Yap in mice and HK-2 cells, we noticed a significant reduction in ZP’s resistance to ferroptosis. These findings suggest that the protective effect of ZP on cisplatin-induced AKI is closely associated with its inhibition of Hippo signaling activation.

## 4. Materials and Methods

### 4.1. Chemicals and Reagents

Dihydrexidine hydrochloride and Cisplatin were purchased from MedChemExpress (Monmouth Junction, NJ, USA). Minimum Essential Medium, penicillin-streptomycin, and fetal bovine serum (FBS) were purchased from Procell (Wuhan, China). Urea assay kit (BUN), creatinine (Cr) assay kit (sarcosine oxidase), superoxide dismutase (SOD), malondialdehyde (MDA), catalase (CAT), and reduced glutathione (GSH) were purchased from Nanjing Jiancheng Bioengineering Institute (Nanjing, China). Kits and probes for MTT, Dihydroethidium (DHE), BCA, and H&E were purchased from Beyotime Biotech (Nantong, China). The following antibodies were used to detect the proteins of interest: Capsase 3 (Cat No. 66470-2-Ig, Proteintech, Rosemont, IL, USA), Capsase 9 (Cat No. 10380-1-AP, Proteintech), Bcl-2 (Cat No. 26593-1-AP, Proteintech), Bax (Cat No. 50599-2-Ig, Proteintech), GPX4 (67763-1-Ig, Proteintech), FSP1 (Cat No. 20886-1-AP, Proteintech), FTH1 (Cat No. 3998S, Cell Signaling), Phospho-YAP (Ser127) (Cat No. 13008S, Cell Signaling), YAP (Cat No. 4912S, Cell Signaling), Phospho-LATS1 (Cat No. 8654S, Cell Signaling), LATS1 (Cat No. 3477S, Cell Signaling), β-Actin (Cat No. 66009-1-Ig, Proteintech), and HRP-conjugated Goat Anti-Rabbit secondary antibody (Cat No. SA00001-2, Proteintech).

### 4.2. Sample Preparation

Based on our previous results (Appendix A) [21], prior to the extraction procedures, hazel leaves were freeze-dried and stored at −80 °C. To create hazel leaf powder, the frozen leaves were crushed. Subsequently, 1 g of the powder was dissolved in a solution consisting of 70% methanol with 1% HCl (1:1 by volume). The solution was then thoroughly mixed, vortexed, and sonicated for 30 min. Following this, the mixture underwent centrifugation at 1800× *g* for 10 min, with the extraction process being repeated twice. The resulting supernatant was combined to form the crude extract, while the residual precipitate was kept at −80 °C. The crude extract was purified from methanol using a vacuum rotary evaporator and subsequently freeze-dried to obtain the extract powder. Phenolic compounds were identified and quantified through the utilization of the HPLC technique (Appendix A). The analysis of phenolic compounds was carried out via the HPLC method (Appendix A). A range of standard phenolic compounds, such as resveratrol, p-Coumaric acid, caffeic acid, chlorogenic acid, gallic acid, luteolin, ellagic acid, kaempferol, and myricetin, were investigated. The quantification of each identified compound in ZP was determined by peak area measurement and is detailed in Table 1. The data revealed that kaempferol had the highest content at 221.9947 mg/g, followed by chlorogenic acid and populin. On the other hand, p-coumaric acid had the lowest content at only 0.0179 mg/g.

### 4.3. Preparation of Animals and Samples Collection

Six-week-old male C57BL/6 mice were acquired from Liaoning Changsheng Technology Co., Ltd., in Liaoning, China. The mice were then randomly allocated into five groups: control (*n* = 6), AKI (20 mg/kg cisplatin in saline, *n* = 6), low-dose ZP group (250 mg/kg ZP in saline, *n* = 6), high-dose ZP group (500 mg/kg, *n* = 6), and ZP+ Dihydrexidine (9 mg/kg Dihydrexidine, i.p.). AKI was induced in the model group by a single intraperitoneal injection of cisplatin. In the ZP and ZP+ Dihydrexidine groups, ZP was orally administered daily, beginning one hour prior to CDDP injection and lasting for 2 days. Following this treatment, the mice were euthanized using amobarbital sodium anesthesia combined with cervical dislocation, and both serum and kidney tissues were collected. All animal procedures adhered to the guidelines outlined in the Guide for the Care and Use of Laboratory Animals (US National Institutes of Health (NIH)), and were specifically approved by the Committee for the Care and Use of Laboratory Animals at Changchun University of Chinese Medicine in Changchun, China. The study protocol was also approved by the Ethics Committee of Changchun University of Chinese Medicine.

### 4.4. Cell Culture

The acquisition of Human proximal tubular epithelial cells (HK-2 cells) was made from Procell (Wuhan, China). Subsequently, the cells were preserved in minimum essential medium (MEM) that was enhanced with 10% fetal bovine serum (FBS) and 1% Penicillin–Streptomycin solution. The incubation of the cells was done at a temperature of 37 °C within an atmosphere of 95% O_2_ and 5% CO_2_, ensuring humidity. The alteration of the medium occurred on a regular basis, specifically every 2–3 days.

### 4.5. Transmission Electron Microscope and Hematoxylin–Eosin Staining (HE)

H&E staining was performed on mice kidney tissues that had been fixed with 4% paraformaldehyde. The stained sections were examined under a light microscope to observe any pathological alterations in mice kidneys. For the ultrastructural analysis, renal tissues from mice were fixed with 2.5% glutaraldehyde and then examined using a transmission electron microscope (TEM; JEM-1400Flash, Tokyo, Japan).

### 4.6. Biochemical Reagent Kit

The biochemical kit from Nanjing Jiancheng Bio-engineering Institute was used to measure levels of urea, creatinine, superoxide dismutase, malondialdehyde, catalase, and reduced glutathione, in accordance with the manufacturer’s guidelines.

### 4.7. ROS Detection

Intracellular ROS levels in cells were determined by in situ probe loading. A diluted DCFH-DA probe solution was substituted, and incubation of the cells persisted for 20 min at 37 °C. Following incubation, the cells were washed and ROS production was visualized and photographed using a fluorescence microscope set to emit light at 488 nm.

### 4.8. TUNEL Staining

Kidney tissues were analyzed using the TdT-mediated dUTP nick-end labeling (TUNEL) in situ cell death detection kit from Roche Diagnostics, following the instructions provided by the manufacturer. The presence of positive cells with green fluorescence was identified under a fluorescent microscope.

### 4.9. Cell Viability Assay

To ensure adherence to the wall, the HK-2 cells were planted in 96-well plates and incubated for 24 h. After that, the cell viability was evaluated using the MTT reagent, following the mentioned cell modeling technique. All steps were carried out strictly according to the provided instructions.

### 4.10. Immunohistochemistry and Immunofluorescence

The renal tissues underwent preservation in 4% paraformaldehyde for 48 h before being embedded in paraffin and sliced into 5 µm sections as per standard procedures. The immunohistochemistry (IHC) and immunofluorescence (IF) techniques were performed in accordance with established methods [40].

### 4.11. Western Blot

The Western blot assay method is consistent with our previous report [41]. Cells or renal tissue were collected and lysed in cell lysis buffer (provided by Beyotime Biotechnology, Nantong, China) while maintaining a low temperature on ice. A BCA protein assay kit was utilized to measure protein levels. After separating protein samples (15 μL) using a 10% SDS-PAGE, they were transferred to a polyvinylidene fluoride membrane. Following this, the membrane was blocked with 5% (*w*/*v*) non-fat milk for 1 h at room temperature. The next step involved overnight incubation of the membranes at 4 °C with the correct primary antibodies. Detection of primary antibody binding was done by exposing the membranes to a secondary antibody linked to horseradish peroxidase for 1 h at room temperature. The bands were visualized by employing the BeyoECL Plus enhanced chemiluminescence kit manufactured by Beyotime Institute of Biotechnology (Nantong, China). The obtained data were analyzed using the ImageJ software (version 1.5.0.26; National Institutes of Health, Bethesda, MD, USA).

### 4.12. Statistical Analysis

The mean ± standard deviations for three independent experiments are presented for all data. Statistical differences were evaluated using one-way analysis of variance or Student’s *t* test. The standard for statistical significance was defined as *p* values < 0.05.

## 5. Conclusions

The present study demonstrated that cisplatin-induced ferroptosis results in damage to renal tubular cells and renal dysfunction, accompanied by an increase in YAP protein phosphorylation. ZP effectively reduces acute kidney injury (AKI) by inhibiting oxidative stress, apoptosis, and ferroptosis in the kidney through inhibiting the Hippo signaling pathway. These findings provide valuable insights into the role of the Hippo pathway in the development of CI-AKI and suggest that ZP could be utilized as a preventive measure for AKI.

## Figures and Tables

**Figure 1 molecules-29-01729-f001:**
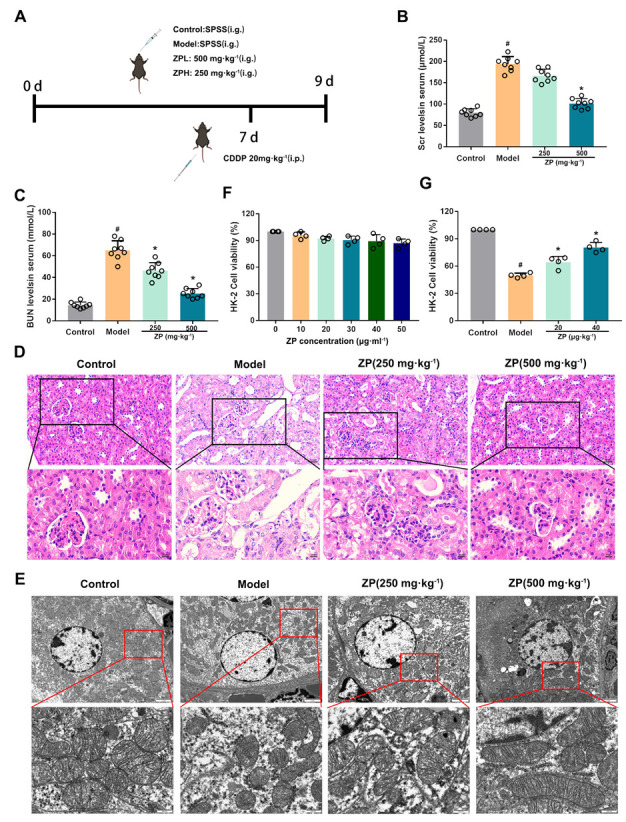
ZP relieves cisplatin-induced acute kidney injury (AKI). (**A**) The schematic design of the animal experiment used in this study. SPSS refers to stroke-physiological saline solution. (**B**,**C**) The levels of blood urea nitrogen (BUN) and serum creatinine (SCr) were measured in mice serum. (**D**,**E**) Histological analysis of kidney tissue was performed using H&E staining and transmission electron microscopy. (**F**,**G**) The proliferation ability of HK-2 cells was evaluated. Results are presented as Mean ± SD. Statistical significance was obtained by one-way ANOVA. # *p* < 0.05 compared with Control group. * *p* < 0.05 compared with Model group.

**Figure 2 molecules-29-01729-f002:**
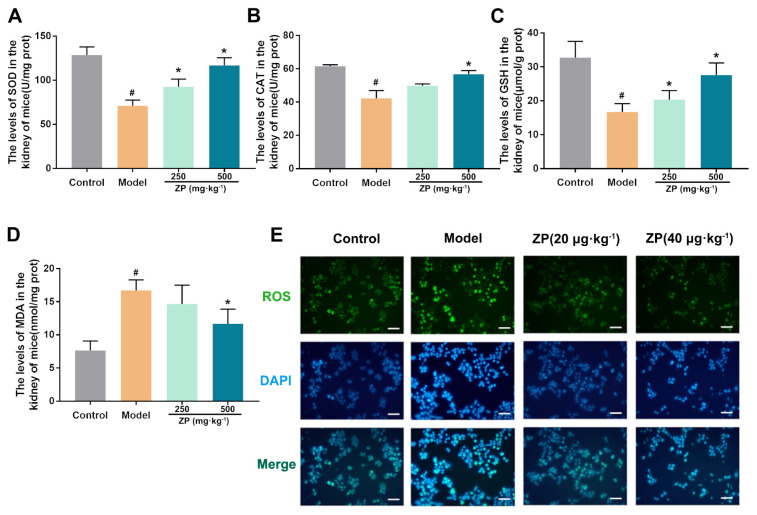
ZP ameliorates oxidative damage in mice kidneys and HK-2 cells. (**A**–**D**) Levels SOD, CAT, GSH, and MDA in mice kidney tissue. (**E**) Representative images of ROS (green) in HK-2 cells. Results are presented as Mean ± SD. Statistical significance was obtained by one-way ANOVA. # *p* < 0.05 compared with Control group. * *p* < 0.05 compared with Model group.

**Figure 3 molecules-29-01729-f003:**
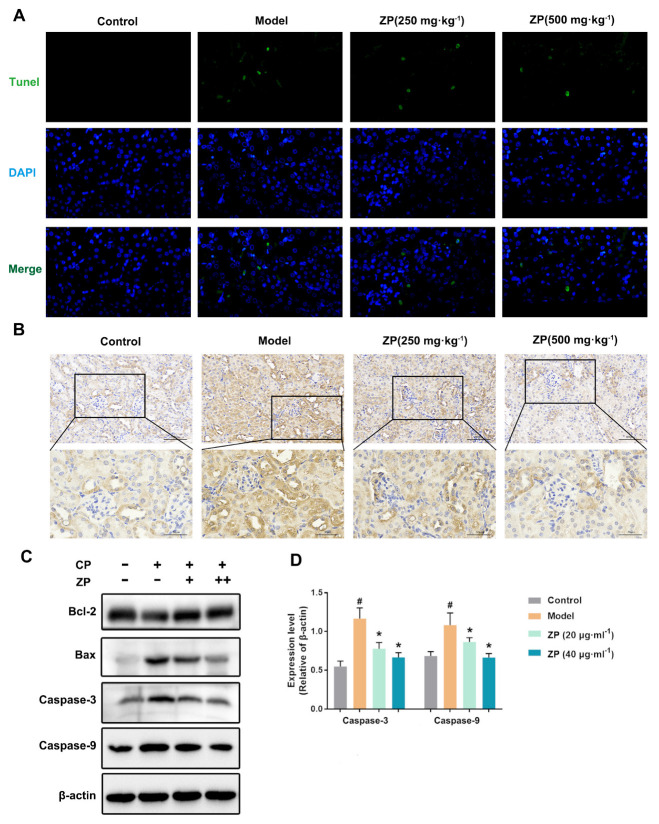
ZP reduces cisplatin-induced apoptosis in mice kidneys and HK-2 cells. (**A**) Representative images of TUNEL staining in mice kidneys. (**B**) Representative images of the protein expression of Caspase-3 in mice kidneys. (**C**,**D**) The protein expression of Bax, Bcl-2, Caspase-3, and Caspase-9 in HK-2 cells (*n* = 3). Results are presented as Mean ± SD. Statistical significance was obtained by one-way ANOVA. # *p* < 0.05 compared with Control group. * *p* < 0.05 compared with Model group.

**Figure 4 molecules-29-01729-f004:**
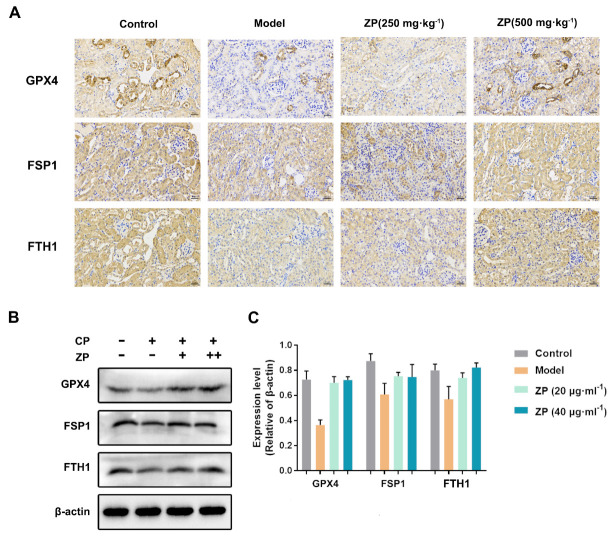
ZP improves cisplatin-induced ferroptosis in mice kidney and HK-2 cells. (**A**) Representative images of GPX4, FSP1, and FTH1 immunohistochemistry in mice kidneys. (**B**,**C**) The protein expressions of GPX4, FSP1, and FTH1 in HK-2 cells (*n* = 3).

**Figure 5 molecules-29-01729-f005:**
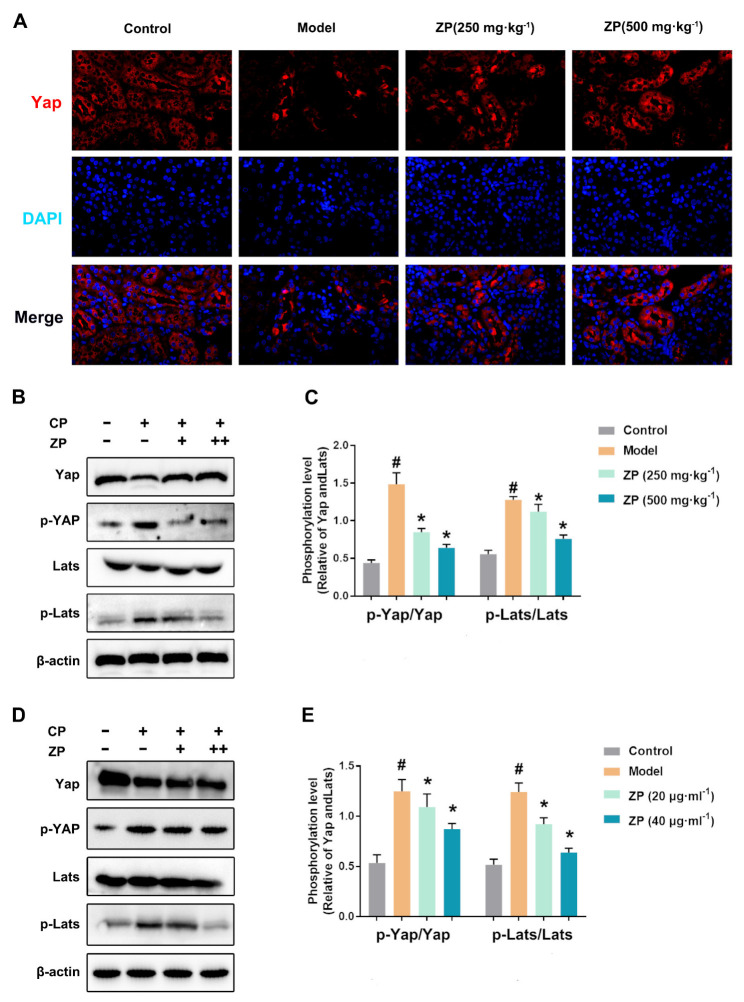
ZP upregulates Hippo signaling in cisplatin-treated mice kidneys and HK-2 cells. (**A**) Representative images of Yap immunofluorescence in mice kidneys. (**B**,**C**) The protein expressions of p-Yap, Yap, p-Lats, and Lats in mice kidneys. (**D**,**E**) The protein expressions of p-Yap, Yap, p-Lats, and Lats in HK-2 cells. Results are presented as Mean ± SD. Statistical significance was obtained by one-way ANOVA. # *p* < 0.05 compared with Control group. * *p* < 0.05 compared with Model group.

**Figure 6 molecules-29-01729-f006:**
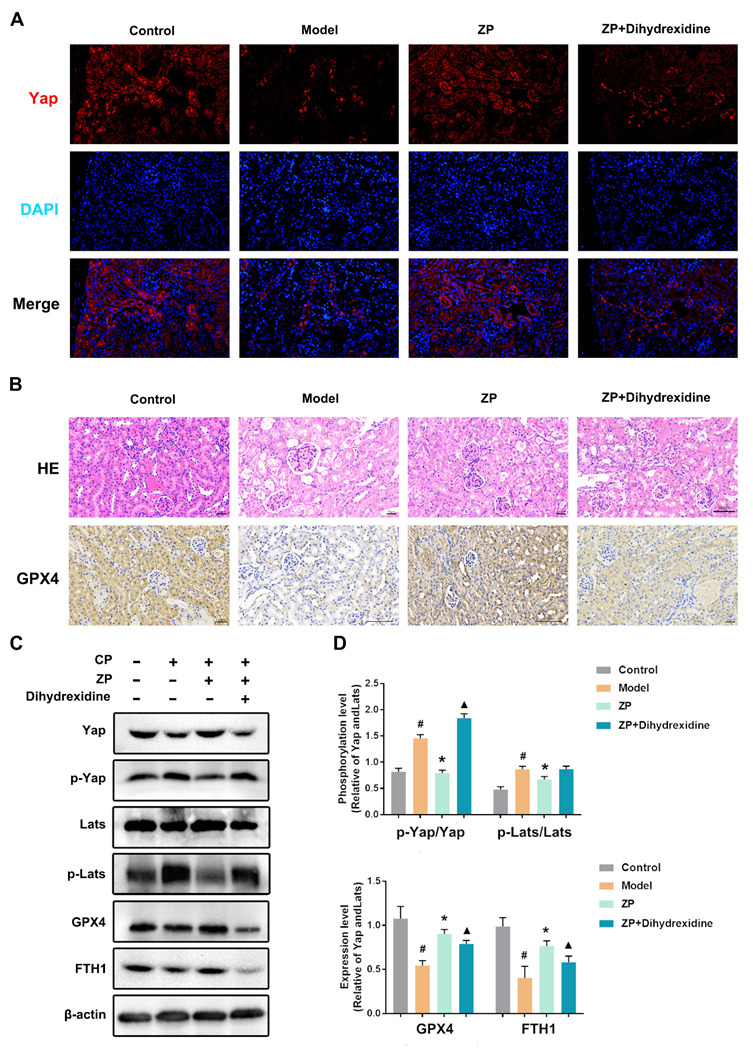
The protective effect of ZP on mice kidneys and HK-2 cells is attenuated after Yap phosphorylation. (**A**) Representative images of Yap immunofluorescence. (**B**) H&E staining and GPX4 immunohistochemistry in mice kidneys. (**C**,**D**) The protein expressions of p-Yap, Yap, p-Lats, Lats, GPX4, and FTH1 in HK-2 cells. Results are presented as Mean ± SD. Statistical significance was obtained by one-way ANOVA. # *p* < 0.05 compared with Control group. * *p* < 0.05 compared with Model group. ▲ *p* < 0.05 compared with ZP group.

**Table 1 molecules-29-01729-t001:** Content of nine phenolic compounds in ZP.

Compounds	Retention Time/min	Concentration of Standard/μg/mL	Quantity Contained/mg/g
Kaempferol	33.688	380	221.995
Chlorogenic acid	12.067	200	8.228
Myricetin	39.546	60	3.956
Ellagic acid	37.246	40	0.564
Luteolin	49.856	50	0.256
Resveratrol	35.988	200	0.214
Caffeic acid	16.262	20	0.111
Gallic acid	3.063	40	0.103
p-Coumaric acid	25.118	35.8	0.018

## Data Availability

Data will be made available on request.

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
