# Peer review of "Hazel Leaf Polyphenol Extract Alleviated Cisplatin-Induced Acute Kidney Injury by Reducing Ferroptosis through Inhibiting Hippo Signaling"

_molecules, 2024, doi:10.3390/molecules29081729_

Round 1
Reviewer 1 Report
Comments and Suggestions for Authors
The authors presented scientifically important results about the mechanism of hazelnut leaf extracts rich in polyphenols on nephroprotection on the molecular level.
The results are clearly presented and conclusions are supported by results.
The originality of the research is not a problem. Still, the authors have to rewrite the manuscript as the plagiarism program detected a high percentage of 49% identity with other available literature sources (the match should not exceed 10 to 20% to may knowledge).
This is the only reason for rejection of the manuscript.
I hope that the authors will carefully rewrite the manuscript using their original sentences in the text and after that re-submit it for new reconsideration for publication.
Kind regards
Reviewer 2 Report
Comments and Suggestions for Authors
The manuscript reported the protective capability of hazel leaf polyphenol extract and investigated the underlying mechanism through inhibiting Hippo signaling. The data is supportive and the overall presentation is clear. A few suggestions might need to be considered:
1. Please check all the acronyms. Please make sure their full name is given at the first appearance. Just a few example: CDDP in line 43, TFRC and FTL in line 69.
2. The paper extensively discussed about ferroptosis and siderosis was mentioned only twice in line 49 and 53. Are they interchangeable concepts? More details are required to be cosistent.
3. Line 116, 123 'HK-2 cells activity' needs to be more specific. Metabolic activity?
4. Please re-write the sentence in line 264 with reference 34. It does not make sense.
5. Based on the reported data herein, AKI leads to the increase in YAP phosphorylation, which inactivates YAP. Thus, Reviewer is confused by the statement in line 289, 'AKI leads to transient YAP/TAZ activation'.
6. In line 336, lysis solution was adjusted to pH 3.0. Is cellulase active at pH 3.0?
7. What is the unit of the last column in Supplemental Table 1?
Round 2
Reviewer 1 Report
Comments and Suggestions for Authors
The author described the protective effect of the extract of hazelnut leaf on cisplatin-induced changes in vitro and in vivo. They found that the Hippo signaling pathway was involved in exerting positive effects of hazelnut leaf extracts on oxidative stress-induced nephrotoxicity of cisplatin.
There are several comments for minor revision
1. Please add a description for used signs for statistical significance in Figures
2. Please check if statistical significance should be added in Figure 4
3. Please delete the catalog numbers of the chemicals
4. The text should be carefully checked for typing mistakes, some of the capital letters should be corrected to the appropriate form
5. The plagiarism percentage is still relatively high try to reduce it to less than 20 percent, but there is still some overlapping that is not deleted from the plagiarism check report.
Kind regards